# Burnout and Suicidal Behaviours in Health Professionals in Portugal: The Moderating Effect of Self-Esteem

**DOI:** 10.3390/ijerph20054325

**Published:** 2023-02-28

**Authors:** Alexandra de Jesus, Liliana Pitacho, Ana Moreira

**Affiliations:** 1Department of Psychology and Sports, Instituto Superior Manuel Teixeira Gomes, 8500-590 Portimão, Portugal; 2Escola Superior Ciências Empresariais, Instituto Politécnico de Setúbal, Campus do IPS—Estefanilha, 2910-761 Setúbal, Portugal; 3Centro de Investigação em Ciências Empresariais (CICE-IPS), 2914-503 Setúbal, Portugal; 4School of Psychology, ISPA—Instituto Universitário, Rua do Jardim do Tabaco 34, 1149-041 Lisboa, Portugal

**Keywords:** burnout, suicidal behaviours, self-esteem, health professionals

## Abstract

The main objective of this study was to investigate the effect of burnout on suicidal behaviours and the mediating effect of self-esteem in this relationship. A total of 1172 healthcare professionals working in Portugal’s private and public sector organisations participated in this study. The results indicate a high level of burnout among these professionals and that exhaustion (β = 0.16; *p* < 0.001) and disengagement (β = 0.24; *p* < 0.001) positively and significantly affect suicidal behaviours. In turn, self-esteem has a significant and negative effect (β = −0.51; *p* < 0.001) on suicidal behaviours. Self-esteem moderates the relationship between disengagement and suicidal behaviours (B = −0.12; *p* < 0.001) and the relationship between exhaustion and suicidal behaviours (B = −0.11; *p* < 0.001), representing an essential variable for future lines of research, namely on the role of self-esteem in preventing burnout and suicidal behaviours in professionals from other professional areas.

## 1. Introduction

Burnout has been identified as an increased occupational risk for several professions, particularly those related to teaching or health care, professions that can promote an important emotional bond or, the opposite, a source of distress. In these professions, it is common for individuals to neglect themselves, putting the needs of others first and/or having a high workload [1].

According to the study by [2], about 21.6% of national health professionals had moderate levels of burnout, and about 47.8% had high levels of burnout. In ten out of twenty districts assessed, about 44% of physicians showed high levels of burnout, and in thirteen districts, 50% of nurses also showed high levels of burnout. Health professionals often find themselves in stressful situations and may develop burnout. Low self-esteem can influence its onset [3].

Suicidal ideation is related to the feeling of helplessness. It occurs more easily in physicians who have several stressful agents simultaneously, leading them to believe that suicide is the only solution left to them. Healthcare professionals may have a higher prevalence of suicide, namely psychiatry and anaesthesiology specialists and internal medicine physicians. The most recurrent method for suicide among healthcare professionals is the consumption of toxic substances due to their easy access and the knowledge of drug effects [4]. However, for [5], poisoning and hanging were the most used methods for suicide by female doctors, while for male doctors, the most commonly used method was firearms.

According to [6], there are high rates of depressive symptoms and intense affective distress in trainees and health professionals, and the relationship between the severity of depression, the intensity of emotional distress and suicidal ideation is particularly strong. To these authors, these professionals and clinical leadership must understand that stress at work is authentic, severe, and prevalent. It is also emphasised that when burnout occurs, one should not stop there, as burnout strongly combines with clinical depression, which is frightening but treatable [6]. For [7], there is a tendency for health professionals to avoid mental health care, a behaviour that may be associated with a social stigma. The objectives of this study were to understand and assess the levels of Burnout, Suicidal behaviours, and Self-esteem in Portuguese Healthcare Professionals, as well as the relationship between these variables. In this way, and based on the objectives and hypotheses, the relevance of this topic is justified given that no studies were found that relate these variables in health care professionals living in Portugal, thus aiming at discussing the results and assessing these mental health indicators. Thus, the following research question was formulated: “Does self-esteem change the relationship between burnout and suicidal behaviours?”.

## 2. Theoretical Framework and Research Hypotheses

### 2.1. Burnout

Burnout is defined as a psychological syndrome that occurs due to high exposure to stressful agents in the work context, conceptualised by high emotional exhaustion (e.g., lethargy, exhaustion, fatigue), depersonalisation (e.g., negative attitudes, irritability, social withdrawal) and reduced personal fulfilment (e.g., reduced productivity and/ or inability to cope with the situation/s) [1].

According to the theoretical model proposed by [8], burnout is a state following constant work experiences in which the subject has low expectations in the existence of positive stimuli and high expectations in the presence of punishment and low expectations about personal skills to control efforts. This level of expectation makes negative feelings predominant in the workplace. In this theoretical component, the aetiology of burnout does not reside only in the subject or in the environment but in an interaction between both. For [9], those who suffer from emotional exhaustion react through depersonalisation, breaking the psychological commitment they maintain in the workplace, presenting a negative self-assessment in personal fulfilment, and leading to burnout. Emotional exhaustion is associated with high levels of depersonalisation and low levels of personal fulfilment. From the perspective of [10], burnout is a cognitive-emotional reaction to chronic stress experienced in the work context. Emotional exhaustion is characterised by the workers’ impersonal perspective and, consequently, by the perceived reduced professional achievement. Nevertheless, the loss of professional identity is visible in those who suffer from depersonalisation, and the reduced self-esteem is reflected in reduced personal fulfilment. It is also referred to as a state derived from the work context. It can occur at the beginning of the working career (e.g., unrealistic expectations), in the middle of the career (e.g., job dissatisfaction) or at the end of the career (e.g., excessive attachment and vocation for work) [11].

Burnout in health professionals is associated with increased rates of depression, substance use, divorce, suicide, medical errors, patient dissatisfaction and unstable interpersonal relationships. This phenomenon damages the individual, family, and organisational sphere, as well as the healthcare system, thanks to the high economic consequences and affects the physician’s well-being, effectiveness, and productivity [12].

According to [13], among healthcare professionals, nurses, younger age groups and women are the ones with the highest levels of burnout. For these authors, one factor that potentiates burnout is the long hours and work. In a study conducted by [14] Vittori et al. (2022), whose participants were anaesthetist/ intensivist physicians, they concluded that the difficulty in describing emotions, feelings, and body sensations (alexithymia) and the distress of symptoms might lead to high levels of burnout. The difficulties experienced by these professionals when trying to get their patients to understand their tasks and the importance of their tasks lead to high scores of alexithymia, psychological symptoms of distress and a decrease in their self-esteem [14]. However, for [15], among the medical specialities with a higher risk of developing burnout are internal medicine, family and general medicine, emergency/urgent medicine, urology and neurology.

### 2.2. Suicidal Behaviour

Suicide is one of the twenty leading causes of death worldwide, killing more than breast cancer [16] and the fourth leading cause of death in subjects aged 15–19 years [17]. Each year, approximately 800,000 individuals die from self-inflicted injury [14], symbolising a huge problematic issue. For every suicide that is accomplished, there is a much higher number of suicide attempts, and this previous attempt consists of the leading risk factor for the suicidal act. In geographical terms, approximately 80% of suicides worldwide occur in underdeveloped and developing countries. In Portugal, only in 2019, 1172 deaths by suicide were recorded, 307 female deaths and 865 male deaths [16]. The most recurrent lethal methods are firearms and pesticide use [17].

Suicide is preventable through the implementation of various strategies, such as limiting access to firearms, developing mental health literacy through the media and/or investing in the socio-emotional development of the subjects), but it is still a serious public health problem [17].

Healthcare professionals are considered a high-risk group for suicide, especially among females, although suicidal acts are generally higher among males [5]. In a study conducted by these authors in the United States of America, they found female doctors complete significantly more suicides than other women who are not doctors. However, although there are more suicides among male doctors than female doctors, there is no significant difference in suicide between male doctors and male non-medical doctors. According to [5], these results are because the suicide rate is also much higher in men in general when compared to women in general. A study by [18] found that the suicide rate among physicians does not differ significantly from the suicide rate in the general population. However, the same is not true for nurses, whose suicide rate is significantly higher than that of the general population. Some factors that may increase the risk of suicidal acts in health professionals, such as the high number of working hours, excessive shifts, improbable routines, easy access to perform the suicidal act, unsatisfactory and negative work environment, lack of support, communication of bad news, and constant contact with the disease, should also be highlighted [19]. In a literature review conducted by [20], they concluded that factors such as depression, anxiety, previous suicide attempt, living alone, and alcohol problems could be antecedents of suicide attempts. However, for these authors, with the COVID-19 Pandemic, new antecedents of suicide attempts emerged, such as economic concerns, considering their working conditions as poor, being afraid of infecting family and friends, and feeling discriminated against or stigmatised by society. Although several authors, including [5], highlight health problems, substance use and psychosocial stressors as risk factors for suicide in physicians, ref. [21] add that the interaction between these factors and the performance of the physician’s role can lead to unique stressors that corrode their well-being and may lead to suicide.

### 2.3. Burnout and Suicidal Behaviour

Concerning health professionals, their low salaries, lack of career progression, and reduced promotion opportunities cause them to have higher anxiety and stress levels [22]. A systematic review conducted from 62 studies from 33 different countries showed that mental health professionals’ prevalence of emotional exhaustion was 40%, 22% for depersonalisation and 18% for a reduced sense of personal accomplishment. A relationship was evidenced between increasing age and depersonalisation levels. Health professionals working in medical inpatient-related settings had lower perceived autonomy at work compared to professionals in community teams [23].

Doctors who have recently completed suicide are up to three times more likely to experience economic problems and up to twice as vulnerable to experiencing health or work problems than those who have not completed suicide [4]. Factors that explain suicidal behaviours in healthcare professionals include work overload, high demands, interpersonal problems and/or workplace harassment. Notwithstanding the discussion of these problems, schedule management and a sense of support in the workplace are protective factors against suicidal behaviours [7]. It should be noted that health professionals, in addition to the clinical management of their patients, are also required to delegate administrative functions. In addition, the reduced ratio between physicians and patients and their high expectation not to make clinical errors are latent, thus representing one of the professions with the highest risk of developing burnout [22]. For [24], burnout in doctors can lead to an increased risk of depression, sleep disorders, fatigue, alcohol and drug abuse, marital dysfunction and, in more severe cases, suicide. This is the reasoning that leads us to formulate the following hypothesis:

**Hypothesis** **1.**
*Burnout significantly and positively affects the frequency of suicidal behaviour.*


### 2.4. Self-Esteem

Self-esteem is defined by an individual’s perception of themselves, directly related to their qualities, skills, and potential [25] and results from a subjective assessment of themselves. Based on this assessment, the subject predisposes, or does not, to his/her acceptance [26].

Self-acceptance is the main characteristic of global self-esteem [27]. This consists of the inner attitude that influences the psychological balance and depends on adaptive processes throughout life. Self-esteem involves two elements: the cognitive, which is represented by the meaning of something specific, such as an idea, material and/or object, and the affective, demonstrated through the intensity of attitudes, whether positive or negative. The concepts that allow understanding self-esteem are appreciation, acceptance, competence, and respect [26].

The theoretical perspective of [28] refers to the understanding of self-esteem as a two-dimensional variable, with an instrumental dimension (self-competence) and another intrinsic dimension (self-sufficiency). The first dimension refers to the value and usefulness of the object itself, such as skills and talents. In contrast, the second represents the object’s qualities, such as morality, where this moral value can be seen in self-esteem as independence or self-sufficiency. Individuals with low self-esteem are shown to lack confidence and efficacy and are critical of themselves [29]. Nevertheless, low self-esteem may prove to be a problem for health professionals given that they engage in a high-pressure work context which, in turn, may lead to the perpetuation of stress and the origin of burnout [30].

The performance of clinical functions is influenced by self-esteem. Nurses with high self-esteem perform better clinical skills and better patient care, while health professionals with low self-esteem may develop less correct behaviours [31].

### 2.5. Self-Esteem and Suicidal Behaviour

The positive association between self-esteem and professional identity presupposes that work is essential for individuals, reinforcing their values. When nurses show a high commitment to their work and feel valued, their self-esteem is strengthened, increasing their perception of competence [30]. There is a relationship between self-esteem and suicidal ideation. Individuals with high self-esteem show adaptive coping strategies to deal with some events compared to those with low self-esteem, who are more sensitive. Low self-esteem predicts and is negatively associated with suicidal ideation [32]. The following hypothesis was formulated:

**Hypothesis** **2.**
*The participants’ self-esteem has a significant and negative association with the frequency of suicidal behaviours.*


### 2.6. Moderating Effect of Self-Esteem

When, for example, nurses have high self-esteem, they are more efficient and flexible in their workplace [33]. Self-esteem plays an important role in well-being and personal satisfaction. Promoting health professionals’ self-esteem increases their quality of life, thus being a protective factor against burnout and suicidal behaviours. A cross-sectional study using a random sample of 306 healthcare professionals in a private hospital in Bangalore, India assessed levels of self-esteem, stress and burnout and concluded that 48.7% of the healthcare professionals showed signs of burnout, 48.4% had high levels of low self-esteem and 38.6% high levels of stress. Participants whose ages were less than 30 years were shown to have lower self-esteem and higher stress levels than participants over 30 years [30]. Finally, the hypothesis was formulated:

**Hypothesis** **3.***Self-esteem has a moderating effect on the relationship between burnout and the frequency of suicidal behaviours*.

The model presented in Figure 1 summarises the research hypotheses formulated in this study.

## 3. Materials and Methods

### 3.1. Data Collection Procedure

This study project was submitted and approved in February 2022 by the Scientific Committee of the master’s degree in Work Psychology and Occupational Health of the Instituto Superior Manuel Teixeira Gomes, Portimão, Portugal. Thus, the request for informed consent and the objectives explained in this study was elaborated on and sent to the authors of the assessment instruments adapted for the Portuguese population to obtain their authorisation for the research. After this, data were collected for all health professionals who so wished, on a voluntary and confidential basis, between February and March 2022, through an online questionnaire shared through social networks, with a 5 to 10 min response time. At the beginning of the questionnaire, participants were informed about this study’s objectives, and their answers’ confidentiality was guaranteed since the analysis of the results would be performed as a whole. We also received support from several entities, such as the Portuguese Association of Cardio pneumologists, the Portuguese Association of Medical Imaging and Radiotherapy, the Portuguese Nurses’ Union and the Portuguese Society of Speech Therapy.

### 3.2. Participants

The sample is composed of 1172 health professionals, 939 female (80.1%) and 233 male (19.9%), aged between 22 and 77 years (M = 40.66), living in Portugal and working in public and private organisations, namely hospitals and health care settings. Regarding marital status, 733 health professionals are married and/or living together (62.5%), 343 are single (29.3%), 80 are divorced (6.8%), 14 are separated (1.2%), and two are widowed (0.2%). As regards the profession, 416 participants are doctors (35.5%), 413 nurses (35.2%), 254 higher technicians in diagnosis and therapy (21.7%) and 89 higher health technicians (7.6%).

Approximately 523 participants answered that their professional activity had lasted more than 17 years (44.6%), 247 indicated between 5 and 10 years (21.1%), 235 participants have worked between 11 and 16 years (20.1%), and 167 answered between 1 and 4 years (14.2%).

Concerning the type of sector in which they perform their functions, 797 participants performed in the public sector (68.0%), 195 in the public and private sector (16.6%) and only 180 in the private sector (15.4%).

Amongst the districts where the health professionals carry out their professional activity, 427 participants are from Porto (36.4%), 234 from Lisbon (20.0%), 119 from Braga (10.2%), 66 from Coimbra (5.6%), 62 from Aveiro (5.3%), 48 from Faro (4.1%), 36 from Setúbal (3.1%), 27 from Viseu (2.3%), 22 from Santarém (1.9%), 20 from Castelo Branco (1.7%), 20 from Leiria (1.7%), 19 from Arquipélago dos Açores (1.6%), 19 from Vila Real (1.6%), 18 from Madeira (1.5%), seven from Beja (0.6%), seven from Bragança (0.6%), seven from Évora (0.6%), six from Viana do Castelo (0.5%), five from Guarda (0.4%) and three from Portalegre (0.3%).

The participants, respectively, for each department are characterised by the following: 243 health professionals belong to the General Medicine department (20.7%), 152 to Primary Health Care (13.0%), 113 to Psychiatry and Mental Health (9.6%), 58 to Internal Medicine and/or Pain Medicine (4.9%), 56 to Surgery and/or Operating Room (4.8%), 47 to Radiology (4.0%), 44 to Physical Medicine and Rehabilitation (3.8%), 40 in Cardiology (3.4%), 38 in Emergency and Urgent Care (3.2%), 38 in Public Health and/or Environmental Health (3.2%), 38 in Pathology and/or Clinical Pathology (3.2%), 32 in Pediatrics and/or Neonatology (2.7%), 28 in Intensive Care Medicine (2.4%), 27 in Pulmonology (2.3%), 25 in Obstetrics and/or Gynecology (2, 1%), 23 of Pharmacy (2.0%), 14 of Oncology (1.2%), 12 of Otorhinolaryngology (1.0%), 12 of Neurology (1.0%), 11 of Clinical Hematology and/or Immunochemotherapy (0.9%), 11 of Dietetics and Nutrition (0.9%), 11 of Ophthalmology (0.9%), 11 of Rheumatology and/or Orthopedics (0.9%), 11 of Nephrology and/or Urology (0.9%), 10 of Dental Medicine (0.9%), 10 Anesthesiology (0.9%), 9 Palliative Care (0.8%), 8 Dermatology (0.7%), 7 Gastroenterology and/or Stomatology (0.6%), 6 Speech and/or Occupational Therapy (0.5%), 6 Endocrinology (0.5%), 5 Management (0.4%), 5 Infectiology (0.4%), 4 Occupational Medicine and/or Occupational Health (0.3%), 2 of Clinical Analyses and Genetics (0.2%), 1 of Immuno-allergology (0.1%), 1 of Nuclear Medicine (0.1%).

### 3.3. Data Analysis Procedure

The first step was to test the metric qualities of the instruments used in this study. After collecting the data and coding them in Microsoft Excel, they were analysed in SPSS Statistics 28 Software (IBM Corp., Armonk, NY, USA) and Amos Graphics 28 (IBM Corp., Armonk, NY, USA). In order to test the validity of the instruments, confirmatory factor analyses were performed for the Oldenburg Burnout Inventory [34] and the Rosenberg Self-Esteem Scale [35]. The procedure was according to a “model generation” logic [36], considering in the analysis of their adjustment, interactively the results obtained: for the chi-square (χ^2^) ≤ 5; for the Tucker Lewis index (TLI) > 0.90; for goodness-of-fit index (GFI) > 0.90; for comparative fit index (CFI) > 0.90; for root mean square error of approximation (RMSEA) ≤ 0.08; root mean square residual (RMSR). A smaller RMSR value corresponds to a better adjustment [37].

For the Suicidal Behaviours Questionnaire Instrument [38], an exploratory factor analysis was performed. We calculated the KMO value, which should be greater than 0.70 [39], and the significance of Bartlett’s sphericity test indicates whether the data come from a multivariate normal population [40].

Internal consistency was tested by calculating Cronbach’s alpha, whose value should vary between “0” and “1”, not assuming negative values [41] and is higher than 0.70, the minimum acceptable in organizational studies [42]. Convergent validity (AVE) and composite reliability were also calculated for each instrument.

The median, asymmetry, kurtosis, minimum and maximum of each item were calculated to test item sensitivity.

The hypotheses formulated in this study were tested by performing simple and multiple linear regressions. Hypothesis 3 (moderation effect) was tested using Macro Process 4.0, developed by [43]. The effect of socio-demographic variables on the variables under study was tested employing mean comparison tests (Student’s *t*-test for independent samples and one-way ANOVA). The association between variables was tested through Pearson’s correlations.

### 3.4. Instruments

The Oldenburg Burnout Inventory [44] is a psychological test that was adapted to the Portuguese population by [34], whose purpose is to assess the Exhaustion and Depersonalisation dimensions [45], where the exhaustion subscale represents the feeling of emptiness, excessive workload, physical, cognitive and emotional exhaustion [46] and the depersonalization subscale reflects the disengagement from the professional environment and attitudes towards work [47] (Appendix A). This test consists of 15 items assessed using a five-point Likert-type scale (1 = Strongly Disagree; 2 = Disagree; 3 = Neither Agree or Disagree; 4 = Agree; 5 = Strongly Agree) [48]. The “disengagement” dimension is composed of items 1, 3, 6, 7, 9, 11, and 15, and the “Exhaustion” dimension is composed of items 2, 4, 5, 8, 10, 12, 14, 16. It should be noted that there are inverted items: 1, 5, 7, 10, 14, and 15. The quotation of the items is performed by calculating the average score of each item. The higher the mean score, the higher the intensity of the dimensions. In order to test the validity of this instrument, a two-factor confirmatory factor analysis was performed. The adjustment indices obtained are adequate (χ^2^/gl = 4.71; GFI = 0.97; CFI = 0.98; TLI = 0.96; RMSEA = 0.056; SRMS = 0.047). As for internal consistency, disengagement presents a Cronbach’s alpha in the value of 0.91 and exhaustion in the value of 0.89. Regarding construct reliability, exhaustion has a value of 0.91 and disengagement has a value of 0.90. Both dimensions showed good convergent validity, with the value of average variance extracted (AVE) of exhaustion being 0.58 and the value of disengagement being 0.53 [49].

The Rosenberg Self-esteem scale [50], adapted for the Portuguese population by [35], is composed of a unidimensional scale with ten items, 5 of positive orientation (1, 3, 4, 7, 10) and 5 of negative orientation (2, 5, 6, 8, 9) (Appendix A). The items range between 1 (strongly disagree) and 4 (strongly agree), where the scores of the negative orientation items are reversed, and the results may vary between 10 and 40 points [27]. A one-factor confirmatory factor analysis was performed to test the validity of this instrument. The adjustment indices obtained are adequate (χ^2^/gl = 3.86; GFI = 0.98; CFI = 0.99; TLI = 0.98; RMSEA = 0.049; SRMS = 0.015). As regards internal consistency, Cronbach’s alpha was 0.92. In turn, construct reliability shows a value of 0.92 and convergent validity a value of AVE of 0.53.

The Suicidal Behaviours Questionnaire-Revised [51] version was adapted for the Portuguese population by [37] (Appendix A). This unidimensional self-report scale consists of four items which aim to assess the frequency and intensity of Suicidal Behaviours, as well as the history of suicidal ideation [30]. In order to test the validity of this instrument, an exploratory factorial was performed, and a KMO value of 0.78 was obtained, which can be considered acceptable [39], and Bartlett’s test of sphericity was significant at *p* < 0.001, being acceptable value to proceed with the analysis, as well as being an indicator that the data come from a multivariate normal population [40]. The factorial structure of this scale was based on a factor which explained 60.6% of the total variability of the scale. All items had factor weights above 0.70. As for internal consistency, Cronbach’s alpha was 0.77.

Regarding the items’ sensitivity, none of the items that make up the instruments grossly violates normality since their absolute values of skewness and kurtosis are below 3 and 7, respectively [52].

## 4. Results

The first step was to perform descriptive statistics of the variables under study and the effect of socio-demographic variables on them, followed by the correlation of variables and hypothesis testing of this research.

### 4.1. Descriptive Statistics of the Variables under Study

In order to understand the answers given by the participants of this study, the descriptive statistics of the variables under study were performed. Regarding self-esteem, the participants of this study were revealed to possess self-esteem (M = 3.19; SD = 0.66) significantly above the central point (2.5) (t (1171) = 35.41; *p* < 0.001; d = 0.93). Concerning disengagement (M = 2.85; SD = 0.97), participants in this study were found to possess levels significantly below the central point (3) (t (1171) = −5.14; *p* < 0.001; d = −0.15). Regarding exhaustion (M = 3.19; SD = 0.89), they were found to have levels significantly above the central point (3) (t (1171) = 7.43; *p* < 0.001; d = 0.16). Concerning suicidal behaviours (M = 1.68; SD = 0.75), participants in this study were found to possess suicidal behaviours significantly below the midpoint of the scale (t (1171) = −82.97; *p* < 0.001). Table 1 below presents the descriptive statistics for each item in the suicidal behaviours scale.

### 4.2. Effect of Socio-Demographic Variables on the Variables under Study

Regarding the relationship between age and the variables under study, younger participants felt higher levels of exhaustion and disengagement and lower levels of self-esteem.

Concerning the effect of gender on the variables under study, statistically, significant differences were only found in the levels of exhaustion (t (1170) = 5.96; *p* < 0.001, d = 0.44) and self-esteem exhaustion (t (1170) = −3.54; *p* < 0.001; d = −0.26). Female participants were found to have significantly higher levels of exhaustion and lower levels of self-esteem than male participants. Although there were no statistically significant differences, female participants showed higher levels of disengagement and fewer suicidal behaviours (Figure 2).

Although the profession does not significantly affect the variables under study, higher education diagnosis and therapy technicians have the highest levels of exhaustion and disengagement and the lowest levels of self-esteem. They are also those who revealed more suicidal behaviours (Figure 3).

Seniority in the job has a significant effect on levels of exhaustion (F (3, 1168) = 8.14; *p* < 0.001), disengagement (F (3, 1168) = 2.81; *p* = 0.039) and self-esteem (F (3, 1168) = 20.43; *p* < 0.001). Participants with higher burnout levels are those with up to 10 years of seniority in the organisation, as opposed to those with more than 17 years of seniority, who show lower levels of burnout. Regarding disengagement, participants with seniority in the organisation between 11 and 16 years are those with the highest levels. Those with a seniority of more than 17 years are those with the lowest levels. Concerning self-esteem, participants with a seniority of more than 17 years are those who showed the highest levels, and those with up to 4 years are those with the lowest levels. The number of suicidal behaviours does not vary significantly according to seniority in the job (Figure 4).

Marital status has a significant effect on exhaustion (F (4, 1167) = 2.92; *p* = 0.020), self-esteem (F (4, 1167) = 7.05; *p* < 0.001) and suicidal behaviours (F (4, 1167) = 2.95; *p* = 0.019). Widowed participants are those with the highest levels of exhaustion but also of self-esteem. Divorced participants have the lowest levels of exhaustion, but on the other hand, they have the highest levels of suicidal behaviours (Figure 5).

Concerning the department where the participant works, the participants who revealed higher levels of exhaustion and disengagement were those who work in the pharmacy department. In turn, these participants were also those who showed the lowest levels of self-esteem. Participants working in the occupational medicine and occupational health department had the lowest levels of exhaustion and disengagement and the highest levels of self-esteem. As for suicidal behaviours, the only participant working in the department of nuclear medicine was the one who revealed having more suicidal behaviours, and those working in the department of clinical analysis and genetics were the ones who had these types of behaviours in smaller amounts (Table 2).

As regards the district of Portugal where they work, as shown in the Table, participants working in the district of Portalegre were the ones who revealed higher levels of exhaustion and disengagement, lower levels of self-esteem and fewer suicidal behaviours. Participants working in the district of Guarda were the ones who revealed lower levels of exhaustion and disengagement and higher levels of self-esteem. The participants who revealed having more suicidal behaviours were those working in the district of Beja (Table 3).

The sector of activity only has a significant effect on disengagement (F (2, 1169) = 3.07; *p* = 0.047. Participants working in the private sector showed the lowest levels of disengagement. It should be noted that participants working in the mixed sector (public and private sector) were the ones who showed more suicidal behaviours (Figure 6).

### 4.3. Study of the Relationship between the Variables

Pearson’s correlations were used to study the relationship between the variables under study.

Suicidal behaviours are positively and significantly associated with exhaustion (r = 0.35; *p* < 0.001) and disengagement (r = 0.32; *p* < 0.001); the higher the levels of exhaustion and disengagement, the higher the frequency of suicidal behaviours (Table 3). The association between suicidal behaviours and self-esteem (r = −0.51; *p* < 0.001) is negative and significant, which indicates that the higher the participants’ self-esteem, the lower the frequency of suicidal behaviours (Table 4).

### 4.4. Hypothesis Test

To test hypothesis 1, a multiple linear regression was performed.

The results indicate to us that disengagement (β = 0.16; *p* < 0.001) and exhaustion (β = 0.24; *p* < 0.001) have a significant and positive effect on the frequency of suicidal behaviours (Table 5). An R^2^_a_ = 0.13 was obtained, which means that the model accounts for 13% of the variability in the frequency of suicidal behaviours. The model is statistically significant (F (2, 1169) = 89.33; *p* < 0.001). Hypothesis one was supported.

To test hypothesis two, a simple linear regression was performed.

The results indicate to us that self-esteem (F (1, 1170) = 412.68; R^2^ = 0.26; β = −0.51; *p* < 0.001) has a significant and negative effect on the frequency of suicidal behaviours (Table 6). The model accounts for 26% of the variability in the frequency of suicidal behaviours. Hypothesis two was supported.

To test hypothesis three, (moderating effect), we used Macro Process 3.5 developed by [35].

The results indicate that self-esteem has a moderating effect both on the relationship between disengagement and the frequency of suicidal behaviours and on the relationship between exhaustion and the frequency of suicidal behaviours (Table 7).

For participants with low levels of self-esteem, when compared to participants with high levels of self-esteem, both disengagement (Figure 7) and exhaustion (Figure 8) become relevant for increasing their frequency of suicidal behaviours. Hypothesis three was supported.

To conclude and better understand the results of this study, we elaborated the Table 8, where the formulated and tested hypotheses are summarised.

## 5. Discussion

This study aimed to study the effect of burnout on the frequency of suicidal behaviour and the moderating effect of self-esteem on this relationship.

As expected, hypothesis one, disengagement and exhaustion significantly and positively affect the frequency of suicidal behaviours. These results are in line with what the literature tells us. According to [4], among the factors that explain suicidal behaviours in health professionals are work overload, high demands, interpersonal problems and/or workplace harassment, which lead to burnout.

Hypothesis two was also confirmed since self-esteem has a significant and negative effect on the occurrence of suicidal behaviours. These results are also in line with the literature since self-esteem is negatively associated with suicidal ideation [32], i.e., the higher the levels of self-esteem, the lower the frequency of suicidal behaviours.

Finally, the moderating effect of self-esteem on the relationship between burnout (disengagement and exhaustion) and suicidal behaviours (H3) was confirmed. In situations of high disengagement and high exhaustion, participants with high levels of self-esteem showed a lower frequency of suicidal behaviours than participants with low self-esteem. In line with [33], promoting health professionals’ self-esteem increases their quality of life, thus being a protective factor against burnout and suicidal behaviours.

The younger participants were the ones who revealed higher levels of exhaustion and disengagement, which is also in line with the literature. For [30], participants aged below 30 years were shown to have lower self-esteem and higher levels of stress when compared to participants aged over 30 years.

It is relevant to justify that the health professionals with the highest levels of disengagement and burnout are the pharmaceutical professionals. This may be due to the COVID-19 Pandemic, which had a significant impact on the profession since all types of associated procedures and personalised care to clients were essential to maintain health care, especially among the elderly, never forgetting that there was always the fear of contagion [53]. Nevertheless, endocrinologists also experience high levels of burnout, and there are several reasons for this, such as [54], the perception of a lack of respect from superiors, excessive bureaucratic burden, and insufficient salaries. In anesthesiologists, the short time, the constant pressure to comply with schedules, the performance of fast procedures, the movement between hospitals, the need to consistently achieve effective results, and the lack of stipulated limits or hierarchical problems are among the factors that can lead to burnout [55]. The Oncology department reflects higher levels of suicidal ideation because these health professionals often deal with patients at the end of their lives, with death, and with all the challenges associated with the profession, such as the excess of patients for each health professional, the bureaucratic and administrative procedures, and the constant update of the profession with frequent scientific advances [56].

Regarding the district of Portugal where they work, participants working in the district of Portalegre were the ones who revealed higher levels of exhaustion and disengagement and lower levels of self-esteem. Contrarily, the participants working in the district of Guarda were revealed to have lower levels of exhaustion and disengagement and higher levels of self-esteem. It should be noted that during the COVID-19 Pandemic, the pressure at the hospital of Portalegre was very high. This district in the interior of Portugal has been struggling with a significant lack of professionals. In what concerns the frequency of suicidal behaviours, professionals working in the district of Beja were the ones who were revealed to have a higher frequency of suicidal behaviours. This district in Portugal is known as the one with the highest suicide rate.

### 5.1. Theoretical and Practical Implications

This research has implications for the theoretical scientific knowledge in the field of burnout and suicidal ideation, contributing to an important mediating factor: self-esteem. Self-esteem is negatively associated with suicidal ideation, a risk factor for suicidal behaviours. The higher the levels of exhaustion and disengagement, the higher the occurrence of suicidal behaviours, and the higher the self-esteem, the lower their occurrence. In this sense, at a practical level, we can infer from this research that the implementation of preventive strategies or programs, such as the promotion of self-esteem, helps to combat suicidal ideation [32], which may be very important for high-risk professions or organisations such as those studied in this research. Healthcare professionals seem to be included in a group where the risk of suicide is high. Among these health professionals, nurses stand out as they have a significantly higher suicide rate than the general population [18]. Other studies indicate that the suicide rate among female physicians is significantly higher than that of non-male physicians, but the same is not true for men [5]. Lately, some studies have emerged, but the scarcity of data on professionals from other areas of health allows understanding this area as a possible target for future research to understand which health professions may be at risk of suicide [22]. Therefore, and from this study, where it was proven that self-esteem moderates the relationship between disengagement and the occurrence of suicidal behaviours, representing an important variable worthy of possible future lines of research in Psychology and mental health, namely in the role of self-esteem in preventing burnout and suicidal behaviours in professionals from other areas.

The limitation and influence at work, the relationship with the patients, the reduced time to accomplish tasks and the bureaucracy have contributed to the burnout of health professionals [57].

In terms of strategies to combat burnout and, consequently, suicidal ideation, we can address not only the work environment but also the personality, attitudes and beliefs of individuals concerning the workplace. To this end, it is necessary to adopt measures at the professional and individual levels to prevent psychological health problems [15]. One possible strategy would be the implementation of more flexible and optimised schedules for each worker [15]. Identifying work stressors, such as the lack of human resources, adopting an action plan to promote well-being at work (e.g., support from managers or a work environment perceived as safe by the employees) and creating a suicide prevention plan in the organisation may be other measures that mitigate the origin of burnout and the occurrence of suicide [58]. According to [24], burnout prevention is more beneficial than treatment. For this author, there is little evidence that treatments such as cognitive behavioural therapy, relaxation, music or creating a positive work environment work after entering burnout.

In the personal sphere, self-care practices and the promotion of self-awareness are imperative for the promotion of mental health, such as seeking professional help (e.g., psychotherapeutic support) and adopting a healthy lifestyle (e.g., regular physical exercise, diversified diet, and a consistent and adequate sleep pattern). The identification and assimilation of a life philosophy that expresses the values and ideals of the individual may also be a protective factor against the development of mental disorders and the balance between personal and professional life, visible, for example, in the identification of the profession that one practice (e.g., if an individual is working in a profession that he or she does not identify with, there will be a conflict at the individual and professional levels, presenting a harmful life philosophy) [15].

Self-care is a relevant variable for a positive mental health indicator, avoiding self-neglect and prioritising protective factors for mental illness, such as prioritising moments with family (e.g., family support), developing hobbies that involve nature or physical exercise (e.g., walking) and relaxation techniques (e.g., meditation) [59]. In turn, ref. [60] suggests that health professionals should practice leisure activities outdoors since they work for long periods in closed environments. Another suggestion from these authors is to participate in group activities outdoors since the knowledge and contact with people with different lives, cultures, and beliefs are very useful in reducing the pressure of work stress. Moderate relaxation and physical exercise help to adequately cope with work stress [61].

### 5.2. Limitations and Future Directions

As a limitation of this study, at the beginning of the study, we only intended to collect data from health professionals working in hospitals; however, health professionals from primary health care adhered to the questionnaire in a relevant way, so we do not exclude them. It should be noted that the pandemic variable was not assessed, which would give us possible indicators of its impact on the psychological functioning of health professionals, as well as the previous existence of mental disorders, which would be important to relate their presence with the levels of suicidal ideation, the number of weekly hours performed during the professional activity, the managerial position and the contractual relationship with the professional entity. Its projection was transversal, assessed in a single moment, not giving a comparative view of two different moments.

Burnout can be intervened through psychological support (e.g., cognitive-behavioural therapy, relaxation techniques, and mental health investment in the institution’s employees). Nevertheless, prevention is the most beneficial investment for both parties, as the treatment and consequences of psychological problems are of high financial and human cost for organisations [24]. From our perspective, this type of research is crucial to understanding the factors that may contribute to burnout and suicidal ideation and assessing the current panorama of the mental health status of health professionals.

## 6. Conclusions

One of the strengths of this study is that we were able to prove that self-esteem has a moderating effect on the relationship between burnout (exhaustion and disengagement) and the frequency of suicidal behaviour. If an individual perceives that they are effective, they will feel a more extraordinary ability to cope with adversity, and self-esteem is essential for individuals’ social and interpersonal integration [31].

The positive and significant effect of burnout on the frequency of suicidal behaviour has been proven. Burnout has negative consequences in the individual and organisational sphere. The number of patients and the number of hours of work, the bureaucratic burden, the conflicts, and the absence of holidays are factors that can influence burnout in physicians, where its high prevalence is associated with productivity loss, medical errors and/or absenteeism [60]. Health professionals who experience several stressors simultaneously are also more likely to have suicidal ideation because, in a situation of helplessness and lack of effective strategies to deal with the situation/s, they may consider suicide the only solution [4].

## Figures and Tables

**Figure 1 ijerph-20-04325-f001:**
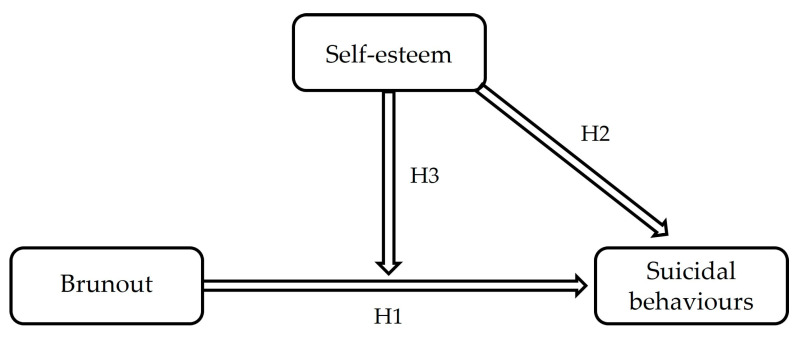
Research Model.

**Figure 2 ijerph-20-04325-f002:**
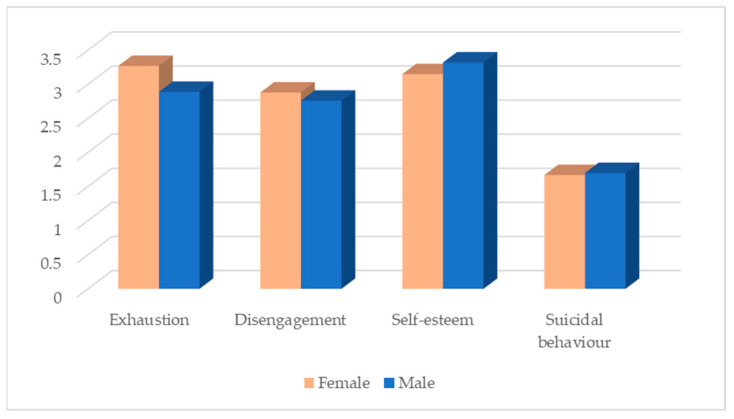
Distribution of the variables under study according to gender.

**Figure 3 ijerph-20-04325-f003:**
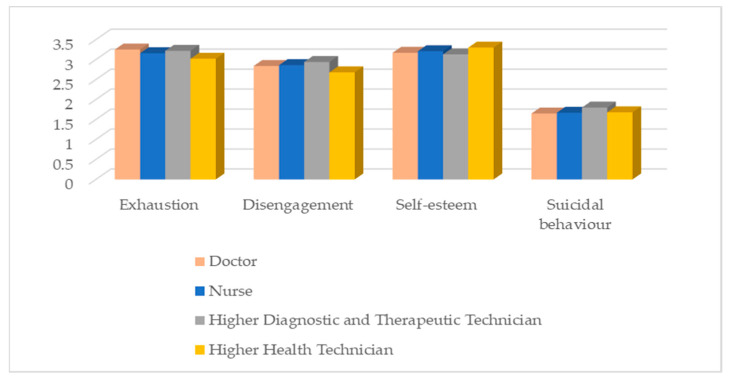
Distribution of the variables under study according to profession.

**Figure 4 ijerph-20-04325-f004:**
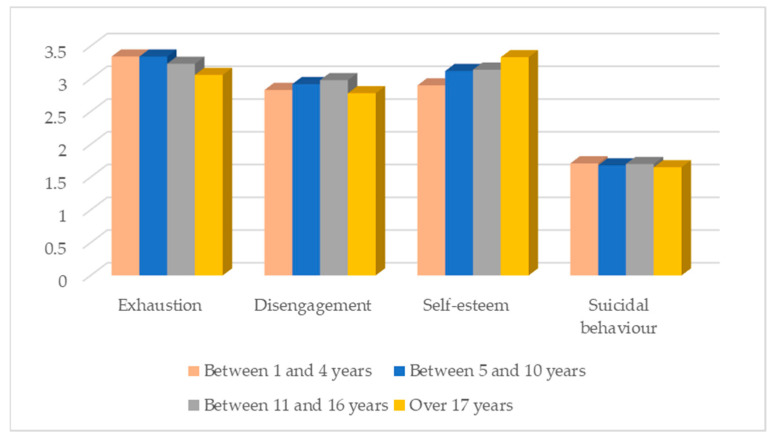
Distribution of the variables under study according to seniority in the job.

**Figure 5 ijerph-20-04325-f005:**
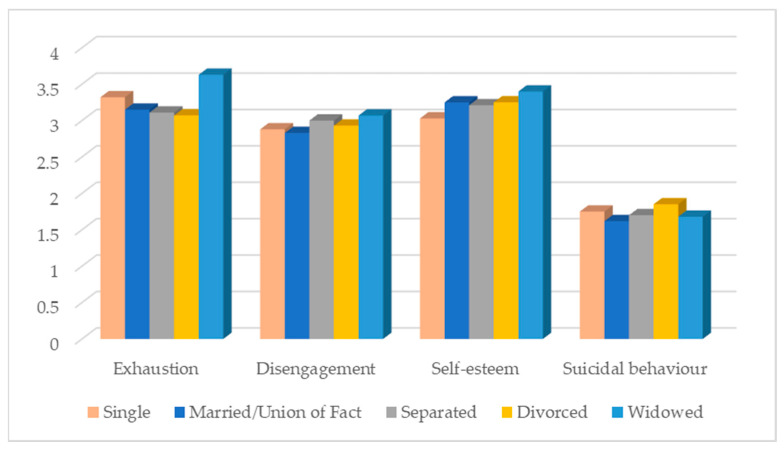
Distribution of the variables under study according to marital status.

**Figure 6 ijerph-20-04325-f006:**
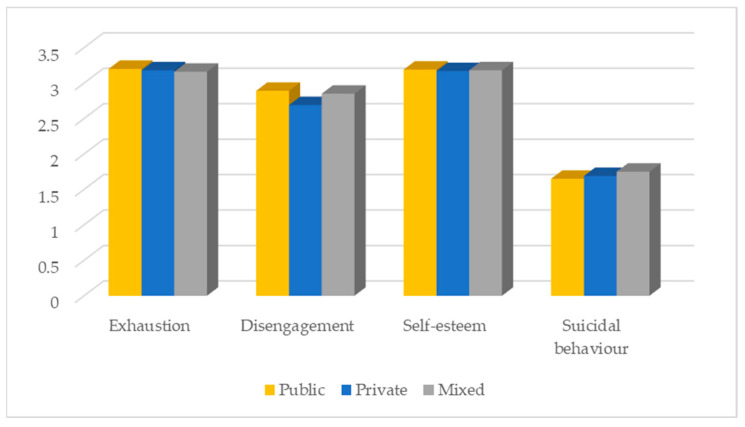
Distribution of the variables under study according to the sector.

**Figure 7 ijerph-20-04325-f007:**
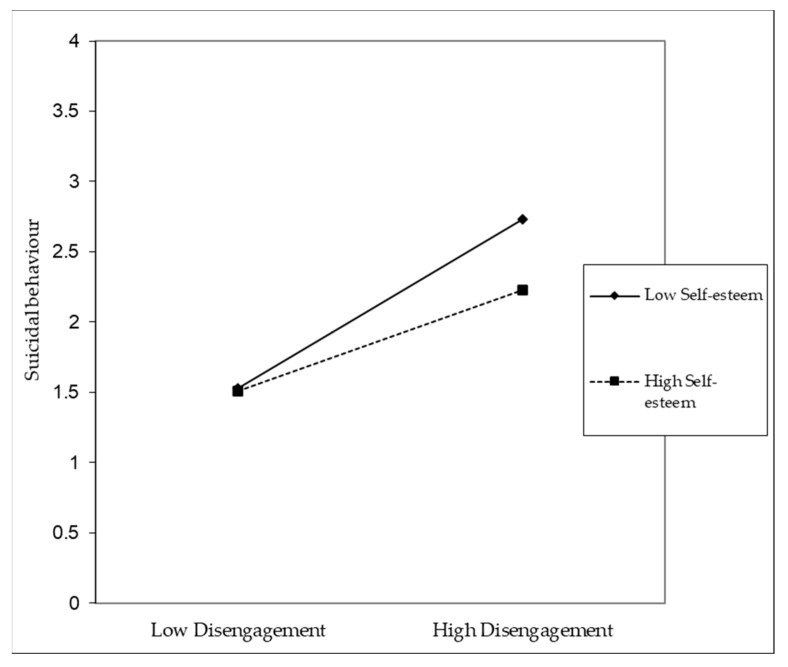
Disengagement × self-esteem interaction effect (B = −0.12; *p* < 0.001).

**Figure 8 ijerph-20-04325-f008:**
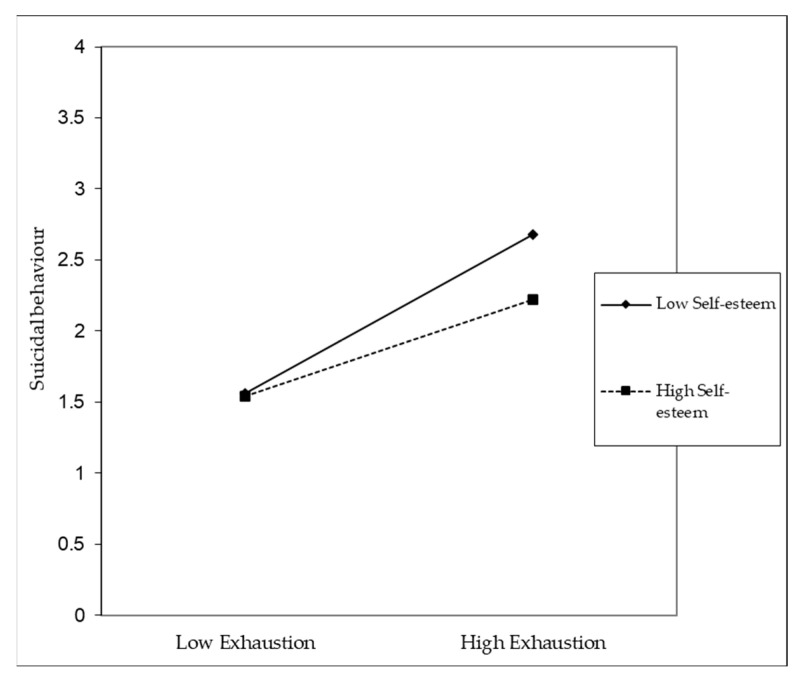
Effect of interaction exhaustion × self-esteem (B = −0.11; *p* < 0.001).

**Table 1 ijerph-20-04325-t001:** Descriptive statistics of each of the items of the suicidal behaviours scale.

Item	Item Scale	Frequency	Percentage
1. Have you ever thought about killing yourself?	Never (1)	723	61.7
	I had just a brief passing thought (2)	320	27.3
	I had only a brief passing thought (3)	73	6.2
	I planned to kill myself at least once, but I didn’t try to do it (4)	30	2.6
	I planned to kill myself at least once and I wanted to die (5)	8	0.7
	I tried to kill myself and I hoped to die (6)	18	1.5
2. How often have you thought of killing yourself in the last year?	Never (1)	908	77.5
	Rarely (1 time) (2)	123	10.5
	Sometimes (2 times) (3)	80	6.8
	Often (3 or 4 times) (4)	28	2.4
	Very often (5 or more times) (5)	33	2.8
3. Have you ever told anyone that you were going to kill yourself or that you might kill yourself?	No (1)	1006	85.8
	Yes, once, but I didn’t want to die (2)	81	6.9
	Yes, once, and I wanted to die (3)	33	2.8
	Yes, more than once, but I didn’t want to do it (4)	32	2.7
	Yes, more than once, and I wanted to do it (5)	20	1.7
4. How likely is it that you might one day attempt suicide?	Never (1)	593	50.6
	Highly unlikely (2)	393	33.5
	Unlikely (3)	127	10.8
	Probable (4)	47	4.0
	Quite likely (5)	8	0.7
	Very likely (6)	4	0.3

**Table 2 ijerph-20-04325-t002:** Distribution of the variables under study according to activity department.

Department (Number of Participants by Department)	*n*	Exhaustion		Disengagement		Self-Esteem		Suicidal Behaviour	
		Mean	SD	Mean	SD	Mean	SD	Mean	SD
Pathological Anatomy/Clinical Pathology	38	2.99	0.87	2.80	0.86	3.18	0.66	1.68	0.69
Anaesthesiology (10)	10	3.44	0.82	2.73	1.24	3.00	0.80	1.78	0.81
Cardiology	40	3.04	0.79	2.56	0.88	3.24	0.54	1.53	0.56
Surgery and/or Operating Theatre	56	3.19	0.93	2.91	0.99	3.21	0.70	1.61	0.63
Palliative Care	9	2.61	0.79	2.00	0.75	3.52	0.58	1.47	0.52
Primary Health Care	152	3.04	0.85	2.74	0.97	3.32	0.58	1.64	0.81
Dermatology	8	3.06	0.55	2.50	0.65	3.46	0.63	1.53	0.53
Dietetics and Nutrition	11	3.24	0.86	2.73	1.00	3.11	0.75	1.64	0.87
Emergency and Urgent Care	38	3.32	0.80	3.18	0.97	3.02	0.80	1.80	0.65
Endocrinology	6	3.67	0.57	3.29	0.90	3.03	0.46	1.50	0.27
Pharmacy	23	3.72	0.80	3.60	0.80	2.89	0.69	1.93	0.93
Gastroenterology/Stomatology	7	2.82	1.18	2.37	1.24	3.39	0.59	1.43	0.37
Management	5	2.95	1.21	2.81	1.44	3.56	0.62	1.65	0.80
Clinical Haematology/Immunohemotherapy	11	3.2	0.74	3.12	0.86	3.35	0.60	1.66	0.69
Infectiology	5	3.40	1.12	3.34	1.21	3.12	0.85	1.70	1.16
Dental Medicine	10	3.04	1.05	2.61	0.75	3.23	0.83	1.63	1.14
Physical and Rehabilitation Medicine	44	3.23	0.91	3.07	1.04	3.07	0.74	1.60	0.71
General and Family Medicine	243	3.36	0.91	3.03	0.96	3.16	0.67	1.64	0.74
Intensive Care Medicine	28	3.14	0.89	2.87	0.97	2.99	0.71	1.71	0.86
Internal Medicine and/or Pain Medicine	58	3.40	0.90	2.94	0.96	3.07	0.73	1.79	0.96
Nephrology and/or Urology	11	2.93	1.00	2.80	1.08	3.25	0.59	1.52	0.43
Neurology	12	3.15	1.01	2.71	0.84	3.24	0.72	1.65	0.81
Obstetrics and/or Gynaecology	25	3.15	0.97	2.54	1.02	3.32	0.56	1.61	0.73
Ophthalmology	11	3.18	0.75	2.57	0.99	3.33	0.79	1.93	0.88
Oncology	14	3.31	0.75	2.78	0.82	2.94	0.66	1.95	1.11
Otolaryngology	12	2.66	0.89	2.65	0.96	3.50	0.59	1.48	0.39
Paediatrics/neonatology	32	3.20	0.78	2.69	0.87	3.08	0.64	1.64	0.67
Pulmonology	27	3.25	0.91	2.72	1.08	3.23	0.64	1.77	0.63
Psychiatry and Mental Health	113	3.05	0.90	2.71	0.95	3.27	0.63	1.83	0.85
Radiology	47	3.41	0.74	2.98	0.88	3.08	0.73	1.68	0.65
Rheumatology/Orthopaedics	11	2.68	0.91	2.55	0.75	3.44	0.43	1.36	0.53
Public and/or Environmental Health	38	3.16	0.95	2.91	0.92	3.15	0.67	1.68	0.55
Speech/Occupational Therapy	6	3.29	0.52	2.52	1.03	2.98	0.50	1.33	0.30

**Table 3 ijerph-20-04325-t003:** Distribution of the variables under study according to Portugal’s district.

Distrito	*n*	Exhaustion		Disengagement		Self-Esteem		Suicidal Behaviour	
		Mean	SD	Mean	SD	Mean	SD	Mean	SD
Aveiro	62	3.45	0.77	3.11	1.01	3.05	0.59	1.64	0.64
Beja	7	3.46	0.71	3.16	1.13	2.96	0.69	1.92	0.85
Braga	119	3.09	0.90	2.77	0.89	3.2	0.67	1.51	0.62
Bragança	7	3.23	1.05	2.94	0.78	3.47	0.43	1.43	0.31
Castelo Branco	20	3.08	0.97	3.32	0.86	3.17	0.67	1.89	0.93
Coimbra	66	3.07	0.80	2.7	0.81	3.2	0.61	1.73	0.81
Évora	7	2.97	0.59	2.57	0.64	3.26	0.56	1.43	0.45
Faro	48	3.17	0.89	2.49	0.89	3.15	0.67	1.66	0.74
Guarda	5	2.55	1.26	2.20	1.32	3.54	0.63	1.55	0.62
Leiria	20	3.21	0.98	3.04	1.01	3.19	0.68	1.78	0.87
Lisboa	234	3.15	0.90	2.76	0.99	3.24	0.64	1.61	0.71
Portalegre	3	3.79	0.19	4.38	0.44	2.90	0.70	1.33	0.38
Porto	427	3.27	0.90	2.96	0.98	3.17	0.67	1.72	0.78
Santarém	22	3.02	0.86	2.86	1.00	3.19	0.77	1.82	0.85
Setúbal	36	3.31	0.78	2.78	0.99	3.17	0.69	1.56	0.63
Viana do Castelo	6	2.83	0.77	2.81	0.82	3.37	0.92	1.83	1.09
Vila Real	19	3.00	1.13	2.66	0.92	3.13	0.66	1.88	1.04
Viseu	27	3.20	0.98	2.75	0.89	3.16	0.85	1.73	0.76
Açores	19	2.94	0.85	2.77	0.79	2.99	0.81	1.88	0.98
Madeira	18	2.86	0.81	2.46	1.14	3.37	0.71	1.82	0.89

**Table 4 ijerph-20-04325-t004:** Pearson’s correlations.

	1	2	3	4
Disengagement	--			
2. Exhaustion	0.68 ***	--		
3. Self-esteem	−0.48 ***	−0.56 ***	--	
4. Suicidal behaviour	0.32 ***	0.35 ***	−0.51 ***	--

Note. *** *p* < 0.001.

**Table 5 ijerph-20-04325-t005:** Multiple linear regression results (H1).

Independent Variables	Dependent Variable	F	*p*	R^2^_a_	β	t	*p*
Disengagement	Suicidal behaviour	89.33 ***	<0.001	0.13	0.16 ***	4.27 ***	<0.001
Exhaustion					0.24 ***	6.43 ***	<0.001

Note. *** *p* < 0.001.

**Table 6 ijerph-20-04325-t006:** Simple linear regression results (H2).

Independent Variable	Dependent Variable	F	*p*	R^2^	β	t	*p*
Self-esteem	Suicidal behaviour	412.68 ***	<0.001	0.26	−0.51 ***	−20.31 ***	<0.001

Note. *** *p* < 0.001.

**Table 7 ijerph-20-04325-t007:** Results of the moderator effect test (H3).

Independent Variables	Suicidal Behaviour (R^2^ = 0.28; *p* < 0.001)			
	B	SE	t	*p*
Constant	1.81	0.35	5.17	<0.001
Disengagement	0.48 ***	0.10	4.82 ***	<0.001
Self-esteem	−0.13	0.10	−1.28	0.200
Disengagement × Self-esteem	−0.12 ***	0.03	−4.16 ***	<0.001
	Suicidal behaviour (R^2^ = 0.16; *p* < 0.001)			
	B	SE	t	*p*
Constant	1.71	0.45	3.78	<0.001
Exhaustion	0.45 ***	0.12	3.87 ***	<0.001
Self-esteem	−0.12	0.13	−0.96	0.338
Exhaustion × Self-esteem	−0.11 ***	0.03	−3.32 ***	<0.001

Note. *** *p* < 0.001.

**Table 8 ijerph-20-04325-t008:** Synthesis of the hypotheses results.

Hypothesis	Decision
Hypothesis 1: Burnout significantly and positively affects the frequency of suicidal behaviour.	Supported
Hypothesis 2: The participants’ self-esteem has a significant and negative association with the frequency of suicidal behaviours.	Supported
Hypothesis 3: Self-esteem has a moderating effect on the relationship between burnout and the frequency of suicidal behaviours.	Supported

## Data Availability

The data presented in this study are available on request from the corresponding author. The data are not publicly available because in their informed consent, participants were informed that the data were confidential and that individual responses would never be known, as data analysis would be of all participants combined.

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
