# Peer review of "Burnout and Suicidal Behaviours in Health Professionals in Portugal: The Moderating Effect of Self-Esteem"

_ijerph, 2023, doi:10.3390/ijerph20054325_

Round 1
Reviewer 1 Report
The study has a clear objective, which is to investigate the mediating effect of self-esteem in the relationships between burnout and suicidal behaviors. The study also has a sufficiently large sample of participants from the health sector. These issues are important but, in my opinion, the authors could greatly improve the manuscript for a publication. My recommendations are:
1. Line 33. “According to the study by [2]”. I think is better: “According to the study by Vicente et al. [2]”. The same in Line 64, Line 70, etc… All around the manuscript
2. Language must be strictly revised.
3. Figure1 doesn't seem appropriate. According to the figure, burnout would depend on self-esteem and suicidal behaviours, but wouldn't it be better to prove that suicidal behaviours depend on burnout and that this relationship is mediated by self-esteem? So it is hypothesized that the more burnout, the more suicidal behaviours, but that self-esteem would act by reducing the value of this relationship. I think the figure does not show the model correctly.
4. The instrument used to measure burnout only measures two dimensions of it. It does not measure personal accomplishment. Why??
5. How convergent validity was evaluated?
6. Line 331. What is mean? Why don’t authors present these results?
7. Lines 404-408. Date exposed in text don’t match with table 3
8. Table4 don’t follow APA norms.
9. Line 419-425. The analysis is the same exposed previously in table 3
1. Authors must comment how they have calculated suicidal behaviour scores. In line 331 authors said that they came to consider suicidal behaviours item by item, but later a global score is considered. Authors must clear that issue.
1. Authors must describe better the instruments that they have used. Is not clear what the suicidal behaviour questionnaire assesses and how. The Items must appear in the manuscript.
1. In my opinion, data could be analysed using better techniques (AMOS SPSS for example).
1. How many participants show suicidal behaviours?
1. Lines 508-516 are not in English.
1. Discussion must be more exhaustive.
1. The study could have more practical implications. For example, authors could indicate the importance of developing organizational strategies to improve the employee’s self-esteem.
Author Response
Article
BURNOUT AND SUICIDAL BEHAVIOURS IN HEALTH PROFESSIONALS IN PORTUGAL: THE MODERATING EFFECT OF SELF-ESTEEM
- Revision 1 -
Dear Reviewer
We appreciate your preliminary comments that will complement our work.
Comment 1: Line 33. “According to the study by [2]”. I think is better: “According to the study by Vicente et al. [2]”. The same in Line 64, Line 70, etc… All around the manuscript.
References have been placed in accordance with this journal. That is why we put them like this.
Comment 2: Language must be strictly revised.
Some sentences have been corrected.
Comment 3: Figure1 doesn't seem appropriate. According to the figure, burnout would depend on self-esteem and suicidal behaviours, but wouldn't it be better to prove that suicidal behaviours depend on burnout and that this relationship is mediated by self-esteem? So it is hypothesized that the more burnout, the more suicidal behaviours, but that self-esteem would act by reducing the value of this relationship. I think the figure does not show the model correctly.
Figure 1 was not correct. It has been corrected.
Comment 4: The instrument used to measure burnout only measures two dimensions of it. It does not measure personal accomplishment. Why??
We used an instrument adapted for the Portuguese population and in its adaptation the instrument has only two dimensions: Exhaustion and Disengagement.
Comment 5: How convergent validity was evaluated?
Convergent validity was analysed by calculating the AVE (Average Variance Extracted) with the data obtained from the confirmatory factor analysis.
Comment 6: Line 331. What is mean? Why don’t authors present these results?
The mean and standard deviation were placed, and the values obtained in the t-test for one sample. The mean and standard deviation and one-sample t-student test results for suicidal behaviours have been added (rows 377 to 381). Table 1 was also added, showing the descriptive statistics for each item.
Comment 7: Lines 404-408. Date exposed in text don’t match with table 3.
Thank you for the correction, as there were errors that have been corrected.
Comment 8: Table4 don’t follow APA norms.
The correction has been made.
Comment 9: Line 419-425. The analysis is the same exposed previously in table 3.
The results in table 3 (now table 4), refer to the results of Pearson's correlations.
Lines 419 to 425 are the results of two simple linear regressions, in which it is natural that the Beta value coincides with the R-value of the Pearson correlations.
Compared to Pearson correlations, the advantage of performing simple linear regression is that we also obtain the coefficient of determination and the significance of the model in linear regression.
Comment 10: Authors must comment how they have calculated suicidal behaviour scores. In line 331 authors said that they came to consider suicidal behaviours item by item, but later a global score is considered. Authors must clear that issue.
It was our mistake, but at this point, we have put the descriptive statistics for each item in table 1.
Comment 11: Authors must describe better the instruments that they have used. Is not clear what the suicidal behaviour questionnaire assesses and how. The Items must appear in the manuscript.
The items of the suicidal behaviours instrument and their descriptive statistics are shown in Table 1.
In addition, appendix A was added, where the instruments used in this study are found.
Comment 12: In my opinion, data could be analysed using better techniques (AMOS SPSS for example).
The factor structure of the burnout and self-esteem scales was tested using confirmatory factor analyses performed with AMOS Graphics software. The data obtained were used to calculate the construct reliability and the average variance extracted.
For the Suicidal behaviours instrument, exploratory factor analysis was performed due to the small number of items (4).
To test the hypotheses 1 and 2, we chose to perform linear regressions and to test the moderation effect, and we used the Macro Process.For the Suicidal behaviours instrument, exploratory factor analysis was performed due to the small number of items (4).
To test the hypotheses 3, we chose to perform linear regressions and to test the moderation effect, and we used the Macro Process.
Comment 13: How many participants show suicidal behaviours?
Table 1 was inserted in the results, where the descriptive statistics for each item are presented, and it can be seen how many participants had suicidal behaviours.
Comment 14: Lines 508-516 are not in English.
It was our lapse. It has been corrected.
Comment 15: The study could have more practical implications. For example, authors could indicate the importance of developing organizational strategies to improve the employee’s self-esteem.
Some practical implications have been added.
A table was added at the end of the results to summarize the confirmed and unconfirmed hypotheses. This makes the results clearer.
We hope to have dealt satisfactorily with the proposed suggestions and made all the requested adjustments, both in form and substance. Yours sincerely,
On behalf of my co-authors,
Reference added to the manuscript:
- Ye G. Y.; Davidson, J. E.; Kim, K.; Zisook, S. Physician death by suicide in the United States: 2012–2016. Journal of Psychiatric Research, 2021, 134:158-165. doi: 10.1016/j.jpsychires.2020.12.064.
- Zisook, S.; Young, I.; Doran, N.; Downs, N.; Hadley, A.; Kirby, B.; McGuire, T.; Moutier, C.; Norcross, W.; Tiamson-Kassab, M. Suicidal Ideation Among Students and Physicians at a U.S. Medical School. OMEGA - Journal of Death and Dying, 2016, 74(1), 35-61 doi:10.1177/0030222815598045
- Alwhaibi, M.; Alhawassi, T.M.; Balkhi, B.; Al Aloola, N.; Almomen, A.A.; Alhossan, A.; Alyousif, S.; Almadi, B.; Bin Essa, M.; Kamal, K.M. Burnout and Depressive Symptoms in Healthcare Professionals: A Cross-Sectional Study in Saudi Arabia. Healthcare, 2022, 10, 2447. https://doi.org/ 10.3390/healthcare10122447
- Vittori, A.; Marinangeli, F.;Bignami, E.G.; Simonini, A.; Vergallo, A.; Fiore, G.; Petrucci, E.; Cascella, M.; Pedone, R. Analysis on Burnout, Job Conditions, Alexithymia, and Other Psychological Symptoms in a Sample of Italian Anesthesiologists and Intensivists, Assessed Just before the COVID-19 Pandemic: An AAROI-EMAC Study. Healthcare, 2022, 10, 1370. https://doi.org/10.3390/healthcare10081370
- Davis MA, Cher BAY, Friese CR, Bynum JPW. Association of US Nurse and Physician Occupation With Risk of Suicide. JAMA Psychiatry, 2021, 14;78(6):1–8. doi: 10.1001/jamapsychiatry.2021.0154.
- García-Iglesias J.J; Gómez-Salgado J.; Fernández-Carrasco F.J.; Rodríguez-Díaz L.; Vázquez-Lara J.M.; Prieto-Callejero B.; Allande-Cussó R. Suicidal ideation and suicide attempts in healthcare professionals during the COVID-19 pandemic: A systematic review. Public Health, 2022, 10:1043216. doi: 10.3389/fpubh.2022.1043216
- Kim, K.; Ye, G.; Haddad, A. M.; Kos, N.; Zisook, S.; Davidson, J. E. Thematic analysis and natural language processing of job‐related problems prior to physician suicide in 2003–2018. Suicide and Life-Threatening Behavior, 2022, 52(5), 1-10. DOI: 10.1111/sltb.12896
- Li, Y.-T.; Chen, S.-J.; Lin, K.-J.; Ku, G.C.-M.; Kao,W.-Y.; Chen, I.-S. Relationships among Healthcare Providers’ Job Demands, Leisure Involvement, Emotional Exhaustion, and Leave Intention under the COVID-19 Pandemic. Healthcare, 2023, 11, 56. https://doi.org/10.3390/healthcare11010056

Reviewer 2 Report
Abstract: Include numerical results in the abstract. Delete the phrase starting with ‘concerning’.
Literature review is incomplete. Important papers on this topic are missing. The literature review does not adequately provide information available on non-physician suicide. Also, in the discussion, conclusion p15, line 499 there are incorrect statements about the risk of suicide in physicians. See authors Ye and separately Davis for studies confirming that in the US physicians are not at higher risk than others.
Search the literature by last name Zisook to find many papers reporting on suicide of healthcare workers in the United States. In summary, nurses were at highest risk.
It is incorrect to state that the pharmacological poisoning is the most common method without other qualifiers. This is true in female nurses and female physicians, but males use firearms or hanging more frequently. Either delete the sentence about method from the introduction or re-read the literature and expand on this to accurately reflect issues of method. Further, it has not been proven that the reason for pharmacological poisoning is related to access. Cases of diversion for the purpose of suicide do happen, but far less often than knowledgeable use of medications. It has been repeatedly stated in the literature specific to healthcare providers that knowledge of how to use medications in a lethal manner may be a stronger risk than access. In either case, the statement is conjecture and has not been derived from evidence. In this section of the paper the word ‘notion’ is not the correct term. Consider changing to ‘knowledge’.
Figures 7 and 8 pls add the statistic to the legend or with an * in the body of the figure.
Table 1: Concerned about presenting this data as raw data, especially with cells of <5 for two reasons: tendency to draw inaccurate conclusions from data that is not ‘proportional’, and 2) confidentiality- potential to link identity with subject. Please rework this table accordingly to account for these concerns.
P3, line 104 not clear what authors are describing about the utility of health literacy in preventing suicide. Please reword for clarity. It is difficult to follow the logic as stated. Consider adding the importance of treating depression as a preventive measure. (With citations).
Please also check the literature for the important missing paper by Zisook which explored burnout and suicidal ideation of healthcare professionals. (2021 or 2022) .
There is a portion of the manuscript that was not translated into English and is still in Portuguese.
Author Response
Article
BURNOUT AND SUICIDAL BEHAVIOURS IN HEALTH PROFESSIONALS IN PORTUGAL: THE MODERATING EFFECT OF SELF-ESTEEM
- Revision 1 -
Dear Reviewer
We appreciate your preliminary comments that will complement our work.
Comment 1: Abstract: Include numerical results in the abstract. Delete the phrase starting with ‘concerning’.
We have accepted your suggestion and made the changes.
Comment 2: Literature review is incomplete. Important papers on this topic are missing. The literature review does not adequately provide information available on non-physician suicide. Also, in the discussion, conclusion p15, line 499 there are incorrect statements about the risk of suicide in physicians. See authors Ye and separately Davis for studies confirming that in the US physicians are not at higher risk than others.
This part has also been reworded, as suggested. Thank you for the papers you have indicated.
Comment 3: Search the literature by last name Zisook to find many papers reporting on suicide of healthcare workers in the United States. In summary, nurses were at highest risk.
We have reworded this part as you suggested by reading the papers by the authors you proposed.
Comment 4: It is incorrect to state that the pharmacological poisoning is the most common method without other qualifiers. This is true in female nurses and female physicians, but males use firearms or hanging more frequently. Either delete the sentence about method from the introduction or re-read the literature and expand on this to accurately reflect issues of method. Further, it has not been proven that the reason for pharmacological poisoning is related to access. Cases of diversion for the purpose of suicide do happen, but far less often than knowledgeable use of medications. It has been repeatedly stated in the literature specific to healthcare providers that knowledge of how to use medications in a lethal manner may be a stronger risk than access. In either case, the statement is conjecture and has not been derived from evidence. In this section of the paper the word ‘notion’ is not the correct term. Consider changing to ‘knowledge’.
As you suggested, we went to read new authors, and this part has been reworded. We hope that the changes made are in line with what you suggested.
Comment 5: Figures 7 and 8 pls add the statistic to the legend or with an * in the body of the figure.
Statistical results have been added to the legend.
Comment 6: Table 1: Concerned about presenting this data as raw data, especially with cells of <5 for two reasons: tendency to draw inaccurate conclusions from data that is not ‘proportional’, and 2) confidentiality- potential to link identity with subject. Please rework this table accordingly to account for these concerns.
All lines with fewer than five participants were eliminated (now table 2).
Comment 7: P3, line 104 not clear what authors are describing about the utility of health literacy in preventing suicide. Please reword for clarity. It is difficult to follow the logic as stated. Consider adding the importance of treating depression as a preventive measure. (With citations).
The sentence has been corrected, as the full stop was misplaced (now lines 124 - 127).
Comment 8: Please also check the literature for the important missing paper by Zisook which explored burnout and suicidal ideation of healthcare professionals. (2021 or 2022).
We have included studies by this author and others in the manuscript. Thank you for the suggestion.
Comment 9: There is a portion of the manuscript that was not translated into English and is still in Portuguese.
It was our lapse. It has been corrected.
A table was added at the end of the results to summarize the confirmed and unconfirmed hypotheses. This makes the results clearer.
We hope to have dealt satisfactorily with the proposed suggestions and made all the requested adjustments, both in form and substance.
Yours sincerely,
On behalf of my co-authors,
Reference added to the manuscript:
- Ye G. Y.; Davidson, J. E.; Kim, K.; Zisook, S. Physician death by suicide in the United States: 2012–2016. Journal of Psychiatric Research, 2021, 134:158-165. doi: 10.1016/j.jpsychires.2020.12.064.
- Zisook, S.; Young, I.; Doran, N.; Downs, N.; Hadley, A.; Kirby, B.; McGuire, T.; Moutier, C.; Norcross, W.; Tiamson-Kassab, M. Suicidal Ideation Among Students and Physicians at a U.S. Medical School. OMEGA - Journal of Death and Dying, 2016, 74(1), 35-61 doi:10.1177/0030222815598045
- Alwhaibi, M.; Alhawassi, T.M.; Balkhi, B.; Al Aloola, N.; Almomen, A.A.; Alhossan, A.; Alyousif, S.; Almadi, B.; Bin Essa, M.; Kamal, K.M. Burnout and Depressive Symptoms in Healthcare Professionals: A Cross-Sectional Study in Saudi Arabia. Healthcare, 2022, 10, 2447. https://doi.org/ 10.3390/healthcare10122447
- Vittori, A.; Marinangeli, F.;Bignami, E.G.; Simonini, A.; Vergallo, A.; Fiore, G.; Petrucci, E.; Cascella, M.; Pedone, R. Analysis on Burnout, Job Conditions, Alexithymia, and Other Psychological Symptoms in a Sample of Italian Anesthesiologists and Intensivists, Assessed Just before the COVID-19 Pandemic: An AAROI-EMAC Study. Healthcare, 2022, 10, 1370. https://doi.org/10.3390/healthcare10081370
- Davis MA, Cher BAY, Friese CR, Bynum JPW. Association of US Nurse and Physician Occupation With Risk of Suicide. JAMA Psychiatry, 2021, 14;78(6):1–8. doi: 10.1001/jamapsychiatry.2021.0154.
- García-Iglesias J.J; Gómez-Salgado J.; Fernández-Carrasco F.J.; Rodríguez-Díaz L.; Vázquez-Lara J.M.; Prieto-Callejero B.; Allande-Cussó R. Suicidal ideation and suicide attempts in healthcare professionals during the COVID-19 pandemic: A systematic review. Public Health, 2022, 10:1043216. doi: 10.3389/fpubh.2022.1043216
- Kim, K.; Ye, G.; Haddad, A. M.; Kos, N.; Zisook, S.; Davidson, J. E. Thematic analysis and natural language processing of job‐related problems prior to physician suicide in 2003–2018. Suicide and Life-Threatening Behavior, 2022, 52(5), 1-10. DOI: 10.1111/sltb.12896
- Li, Y.-T.; Chen, S.-J.; Lin, K.-J.; Ku, G.C.-M.; Kao,W.-Y.; Chen, I.-S. Relationships among Healthcare Providers’ Job Demands, Leisure Involvement, Emotional Exhaustion, and Leave Intention under the COVID-19 Pandemic. Healthcare, 2023, 11, 56. https://doi.org/10.3390/healthcare11010056

Reviewer 3 Report
Thank you for submitting the manuscript. I have read with great interest and attention your paper which appears solid, well written and easy to read. However, there are some revisions which in my opinion are necessary to increase the quality of the manuscript.
Please specifically cite the author of reference 2 in the body of the introduction.
Please specifically cite the author of reference 6 in the body of the introduction
Please specifically cite the author of reference 7 in the body of the introduction (Please explicitly cite the names of the authors of the references you explicitly cite: eg .... as demonstrated by Caius, or postulated by Sempronius, not just the number of the references.)
The sentence "Emotional exhaustion is associated with high levels of depersonalisation and low levels of personal fulfilment, while depersonalisation is related to the inability to cope with emotional ex
haustion" does not seem clear to me. In fact, the three dimensions of burnout are mixed, which are actually independent. Please emulate the sentence, and in this regard I invite you to read the following references and use them in your bibliography: doi: 10.3390/healthcare11010056. doi: 10.3390/healthcare10081370. doi: 10.3390/healthcare10122447.
Lines 107-110: please provide refrrences.
Lines 508-516: even if Portuguese is a beautiful language, the manuscript must be written in English. Please translate.
I hope these comments are helpful to you.
Kind Regards
Author Response
Article
BURNOUT AND SUICIDAL BEHAVIOURS IN HEALTH PROFESSIONALS IN PORTUGAL: THE MODERATING EFFECT OF SELF-ESTEEM
- Revision 1 -
Dear Reviewer
We appreciate your preliminary comments that will complement our work.
Comment 1: Please specifically cite the author of reference 2 in the body of the introduction.
Please specifically cite the author of reference 6 in the body of the introduction
Please specifically cite the author of reference 7 in the body of the introduction (Please explicitly cite the names of the authors of the references you explicitly cite: eg .... as demonstrated by Caius, or postulated by Sempronius, not just the number of the references.)
References have been placed in accordance with this journal. That is why we put them like this.
Comment 2: The sentence "Emotional exhaustion is associated with high levels of depersonalisation and low levels of personal fulfilment, while depersonalisation is related to the inability to cope with emotional exhaustion" does not seem clear to me. In fact, the three dimensions of burnout are mixed, which are actually independent. Please emulate the sentence, and in this regard I invite you to read the following references and use them in your bibliography: doi: 10.3390/healthcare11010056. doi: 10.3390/healthcare10081370. doi: 10.3390/healthcare10122447.
We have read all the articles you have suggested for improving the manuscript, for which I thank you very much. The changes that we considered important have been made.
Comment 3: Lines 107-110: please provide references.
The text has been reworded between former lines 107 and 114, and references have been included (now lines 124-127).
Comment 4: Lines 508-516: even if Portuguese is a beautiful language, the manuscript must be written in English. Please translate.
It was our lapse. It has been corrected.
A table was added at the end of the results to summarize the confirmed and unconfirmed hypotheses. This makes the results clearer.
We hope to have dealt satisfactorily with the proposed suggestions and made all the requested adjustments, both in form and substance. Yours sincerely,
On behalf of my co-authors,
Reference added to the manuscript:
- Ye G. Y.; Davidson, J. E.; Kim, K.; Zisook, S. Physician death by suicide in the United States: 2012–2016. Journal of Psychiatric Research, 2021, 134:158-165. doi: 10.1016/j.jpsychires.2020.12.064.
- Zisook, S.; Young, I.; Doran, N.; Downs, N.; Hadley, A.; Kirby, B.; McGuire, T.; Moutier, C.; Norcross, W.; Tiamson-Kassab, M. Suicidal Ideation Among Students and Physicians at a U.S. Medical School. OMEGA - Journal of Death and Dying, 2016, 74(1), 35-61 doi:10.1177/0030222815598045
- Alwhaibi, M.; Alhawassi, T.M.; Balkhi, B.; Al Aloola, N.; Almomen, A.A.; Alhossan, A.; Alyousif, S.; Almadi, B.; Bin Essa, M.; Kamal, K.M. Burnout and Depressive Symptoms in Healthcare Professionals: A Cross-Sectional Study in Saudi Arabia. Healthcare, 2022, 10, 2447. https://doi.org/ 10.3390/healthcare10122447
- Vittori, A.; Marinangeli, F.;Bignami, E.G.; Simonini, A.; Vergallo, A.; Fiore, G.; Petrucci, E.; Cascella, M.; Pedone, R. Analysis on Burnout, Job Conditions, Alexithymia, and Other Psychological Symptoms in a Sample of Italian Anesthesiologists and Intensivists, Assessed Just before the COVID-19 Pandemic: An AAROI-EMAC Study. Healthcare, 2022, 10, 1370. https://doi.org/10.3390/healthcare10081370
- Davis MA, Cher BAY, Friese CR, Bynum JPW. Association of US Nurse and Physician Occupation With Risk of Suicide. JAMA Psychiatry, 2021, 14;78(6):1–8. doi: 10.1001/jamapsychiatry.2021.0154.
- García-Iglesias J.J; Gómez-Salgado J.; Fernández-Carrasco F.J.; Rodríguez-Díaz L.; Vázquez-Lara J.M.; Prieto-Callejero B.; Allande-Cussó R. Suicidal ideation and suicide attempts in healthcare professionals during the COVID-19 pandemic: A systematic review. Public Health, 2022, 10:1043216. doi: 10.3389/fpubh.2022.1043216
- Kim, K.; Ye, G.; Haddad, A. M.; Kos, N.; Zisook, S.; Davidson, J. E. Thematic analysis and natural language processing of job‐related problems prior to physician suicide in 2003–2018. Suicide and Life-Threatening Behavior, 2022, 52(5), 1-10. DOI: 10.1111/sltb.12896
- Li, Y.-T.; Chen, S.-J.; Lin, K.-J.; Ku, G.C.-M.; Kao,W.-Y.; Chen, I.-S. Relationships among Healthcare Providers’ Job Demands, Leisure Involvement, Emotional Exhaustion, and Leave Intention under the COVID-19 Pandemic. Healthcare, 2023, 11, 56. https://doi.org/10.3390/healthcare11010056

Round 2
Reviewer 1 Report
This reviewer was impressed with the authors’ diligence in responding to this reviewer’s comments and questions. Most of this reviewer’s comments and questions were successfully addressed, but several outstanding points remain. However, most of the remaining comments and questions are relatively minor, and assuming the authors can successfully them, this article can be published. Specifically, there remains the following comments:
1. As authors have evaluated convergent validity by calculating the AVE (Average Variance Extracted) with the data obtained from the confirmatory factor analysis, I think that they have to mentioned this in the manuscript for all instruments. For example: Both dimensions showed a good convergent validity, with the value of Average Variance Extracted (AVE) of exhaustion being 326 0.58 and the value of the distance being 0.53 [49].
2. In Table 2 file 4, column for SD of suicidal behaviours instead .625 must 0.62. As in Table 3, SD in Table 2 have to begin with 0. in all cases.
3. Authors don’t reply to my concern about how they have calculated suicidal behaviour scores. In Figures and Tables, an only score for the variable “suicidal behaviours” appear. Authors must explain if that suicidal behavior score from which they then calculate the average, SD, etc. it is the sum of the responses to the four items of the questionnaire, the sum of the behaviors with scores greater than 3, the mean of the responses.....??
Author Response
Article
BURNOUT AND SUICIDAL BEHAVIOURS IN HEALTH PROFESSIONALS IN PORTUGAL: THE MODERATING EFFECT OF SELF-ESTEEM
- Revision 2 -
Dear Reviewer
We appreciate your preliminary comments that will complement our work.
Comment 1: As authors have evaluated convergent validity by calculating the AVE (Average Variance Extracted) with the data obtained from the confirmatory factor analysis, I think that they have to mentioned this in the manuscript for all instruments. For example: Both dimensions showed a good convergent validity, with the value of Average Variance Extracted (AVE) of exhaustion being 0.58 and the value of the distance being 0.53 [49].
The correction you suggested has been made.
Comment 2: In Table 2 file 4, column for SD of suicidal behaviours instead .625 must 0.62. As in Table 3, SD in Table 2 have to begin with 0. in all cases.
The correction has been made.
Comment 3: Authors don’t reply to my concern about how they have calculated suicidal behaviour scores. In Figures and Tables, an only score for the variable “suicidal behaviours” appear. Authors must explain if that suicidal behavior score from which they then calculate the average, SD, etc. it is the sum of the responses to the four items of the questionnaire, the sum of the behaviors with scores greater than 3, the mean of the responses.....??
Thank you for the question. The average score of the 4 items that make up the instrument has been calculated.
We hope to have dealt satisfactorily with the proposed suggestions and made all the requested adjustments, both in form and substance. Yours sincerely,
On behalf of my co-authors,
Reference added to the manuscript:
- Zisook, S.; Doran, N.: Mortali, M.; Hoffman, L.; Downs, N.; Davidson, J; Fergerson, B.; Rubanovich, C.K.; Shapiro, D.; Tai-Seale, M.; Iglewicz, A.; Nestsiarovich, A.; Moutier, C.Y. Relationship between burnout and Major Depressive Disorder in health professionals: A HEAR report. Journal of Affective Disorders, 2022, 312, 259-267. doi: 10.1016/j.jad.2022.06.047. Epub 2022 Jun 24. PMID: 35760197
Reference removed from the manuscript:
- Zisook, S.; Young, I.; Doran, N.; Downs, N.; Hadley, A.; Kirby, B.; McGuire, T.; Moutier, C.; Norcross, W.; Tiamson-Kassab, M. Suicidal Ideation Among Students and Physicians at a U.S. Medical School. OMEGA - Journal of Death and Dying, 2016, 74(1), 35-61 doi:10.1177/0030222815598045

Reviewer 2 Report
Authors have not yet corrected the citation regarding burnout. Reference 6 could be replaced with this more current manuscript. The key point of the findings in this paper are that approximately 1/3 of the clinicians in the study had both burnout and depression, meaning that it would be dangerous to screen for burnoout in isolation of depression as depression warrants clinical treatment, and undetected or treated could lead to suicide. https://www.sciencedirect.com/science/article/pii/S0165032722007078?casa_token=3-zXsh1eZvgAAAAA:UJq0DO85Gkos_ILYonRaYUQRTM596mwu0gUJKCmugG6SV-74i0vdZGAAn9mDSoRTfPvWfZy4cA
Also, The new sentences have English grammar issues. It is unclear when the word 'author' is used in the sentence what the author is referring to.
The word 'commit' needs to be changed to the more acceptable term of 'complete' or 'died by suicide'. The term 'commit' is not used because it has a criminal (vs. disease-focused) connotation.
Author Response
Article
BURNOUT AND SUICIDAL BEHAVIOURS IN HEALTH PROFESSIONALS IN PORTUGAL: THE MODERATING EFFECT OF SELF-ESTEEM
- Revision 1 -
Dear Reviewer
We appreciate your preliminary comments that will complement our work.
Comment 1: Authors have not yet corrected the citation regarding burnout. Reference 6 could be replaced with this more current manuscript. The key point of the findings in this paper are that approximately 1/3 of the clinicians in the study had both burnout and depression, meaning that it would be dangerous to screen for burnout in isolation of depression as depression warrants clinical treatment, and undetected or treated could lead to suicide.
https://www.sciencedirect.com/science/article/pii/S0165032722007078?casa_token=3-zXsh1eZvgAAAAA:UJq0DO85Gkos_ILYonRaYUQRTM596mwu0gUJKCmugG6SV-74i0vdZGAAn9mDSoRTfPvWfZy4cA
The proposed change has been made and the reference updated. Thank you for the paper you have indicated.
Comment 2: Also, the new sentences have English grammar issues. It is unclear when the word 'author' is used in the sentence what the author is referring to.
The word author was removed in the situations you mentioned.
Comment 3: The word 'commit' needs to be changed to the more acceptable term of 'complete' or 'died by suicide'. The term 'commit' is not used because it has a criminal (vs. disease-focused) connotation.
Thank you for the suggestion. The substitution has been effectuated.
We hope to have dealt satisfactorily with the proposed suggestions and made all the requested adjustments, both in form and substance.
Yours sincerely,
On behalf of my co-authors,
Reference added to the manuscript:
- Zisook, S.; Doran, N.: Mortali, M.; Hoffman, L.; Downs, N.; Davidson, J; Fergerson, B.; Rubanovich, C.K.; Shapiro, D.; Tai-Seale, M.; Iglewicz, A.; Nestsiarovich, A.; Moutier, C.Y. Relationship between burnout and Major Depressive Disorder in health professionals: A HEAR report. Journal of Affective Disorders, 2022, 312, 259-267. doi: 10.1016/j.jad.2022.06.047. Epub 2022 Jun 24. PMID: 35760197
Reference removed from the manuscript:
- Zisook, S.; Young, I.; Doran, N.; Downs, N.; Hadley, A.; Kirby, B.; McGuire, T.; Moutier, C.; Norcross, W.; Tiamson-Kassab, M. Suicidal Ideation Among Students and Physicians at a U.S. Medical School. OMEGA - Journal of Death and Dying, 2016, 74(1), 35-61 doi:10.1177/0030222815598045
